# Expression Patterns of Drosophila Melanogaster Glutathione Transferases

**DOI:** 10.3390/insects13070612

**Published:** 2022-07-07

**Authors:** Elodie Gonis, Stéphane Fraichard, Thomas Chertemps, Arnaud Hecker, Mathieu Schwartz, Francis Canon, Fabrice Neiers

**Affiliations:** 1Centre des Sciences du Goût et de l’Alimentation, Université de Bourgogne Franche-Comté, INRAE, CNRS, Institut Agro, 21000 Dijon, France; elodiegonis@gmail.com (E.G.); stephane.fraichard@u-bourgogne.fr (S.F.); mathieu.schwartz@inrae.fr (M.S.); francis.canon@inrae.fr (F.C.); 2Institut d’Ecologie et des Sciences de l’Environnement de Paris, Sorbonne Université, INRAE, CNRS, IRD, UPEC, 75005 Paris, France; thomas.chertemps@upmc.fr; 3IAM, Université de Lorraine, INRAE, 54000 Nancy, France; arnaud.hecker@univ-lorraine.fr

**Keywords:** glutathione transferases, expression, *Drosophila melanogaster*, function, localization, insecticide resistance

## Abstract

**Simple Summary:**

Glutathione transferases are key enzymes found in all living species. In insects, two classes of these enzymes are specific: the Delta and Epsilon classes. These classes of glutathione transferases are generally considered supports during the evolution for the adaptability of the species of an insect. We measured the expression levels of the 25 enzymes forming these two classes in different organs of *Drosophila melanogaster*. Some glutathione transferases are tissue-specific, others are specifically underexpressed in some tissues, and some appear expressed at the same level in all the tested tissues. All glutathione transferases present a specific pattern that does not depend on their sequence proximity. This study allows us to better understand the potential role of these enzymes by analyzing their expression profile and evolutionary links.

**Abstract:**

Glutathione transferases (GSTs) are ubiquitous enzymes that catalyze the conjugation of glutathione to various molecules. Among the 42 GSTs identified in *Drosophila melanogaster*, Delta and Epsilon are the largest classes, with 25 members. The Delta and Epsilon classes are involved in different functions, such as insecticide resistance and ecdysone biosynthesis. The insect GST number variability is due mainly to these classes. Thus, they are generally considered supports during the evolution for the adaptability of the insect species. To explore the link between Delta and Epsilon GST and their evolution, we analyzed the sequences using bioinformatic tools. Subgroups appear within the Delta and Epsilon GSTs with different levels of diversification. The diversification also appears in the sequences showing differences in the active site. Additionally, amino acids essential for structural stability or dimerization appear conserved in all GSTs. Quantitative real-time polymerase chain reaction (qRT-PCR) analysis revealed that the transcripts corresponding to these two classes are heterogeneously expressed within *D. melanogaster*. Some GSTs, such as GSTD1, are highly expressed in all tissues, suggesting their general function in detoxification. Conversely, some others, such as GSTD11 or GSTE4, are specifically expressed at a high level specifically in antennae, suggesting a potential role in olfaction.

## 1. Introduction

Glutathione transferases (GST) are ubiquitous enzymes that are present in all kingdoms of life. They represent a key family of enzymes involved in various processes, including detoxification. GST members constitute the GSTome [1] and can be divided into three superfamilies based on their cellular localization: mitochondrial, microsomal and cytosolic, also named canonical GSTs. The mitochondrial class (Kappa GSTs) is present in many eukaryotes but is absent in insects [2]. Only one microsomal GST (also called membrane-associated protein in eicosanoid and glutathione metabolism, or MAPEG) was identified in *Drosophila melanogaster* [3]. The microsomal class is not evolutionarily related to the cytosolic classes. Their respective sequences are different, but the two classes share a common transferase activity. Consequently, all cytosolic GSTs evolved from a common ancestor. The classes are designated by Greek letters; generally, a class contains several GSTs, which are numbered using Arabic numerals. In *D. melanogaster*, 42 GSTs belonging to six different classes have been identified: Theta, Omega, Sigma, Zeta, Delta, and Epsilon. The Delta and Epsilon classes appear to be specific to arthropods, including insects and, thus, *D. melanogaster*. All solved structures of Delta and Epsilon GSTs are homodimeric, similar to most GSTs. The GST numbers are variable among the different insect species [4]. This variability is due mainly to the number of Delta and Epsilon classes. For instance, *Apis mellifera* has only one GST member in the Delta class and not any GST member in the Epsilon class. In contrast, more than half of the GSTs belong to these two classes in *D. melanogaster* or *Anopheles gambiae*. This variation has been proposed to play a key role in the adaptation of insects in response to environmental selection pressures [4]. Among the 42 distinct GSTs identified in *D. melanogaster*, 25 belong to the Delta and Epsilon classes (GSTD and GSTE, respectively), representing the largest classes. All 25 GSTDs and GSTEs present glutathione transferase activity and can potentially be involved in the detoxification process; however, only GSTD3 does not exhibit this activity [5,6].

GSTs are known to belong to the phase II enzymes of the general detoxification process. This process can be divided into three phases, including a first phase with enzymes involved in the activation of exogenous molecules, and a second phase where different enzymes can act, including GSTs, which is dedicated to the enzymatic addition of a polar group. The third and last phase occurs when the first two phases act inside the cells and consist of the elimination of the polar molecule formed from the cell. Drosophila GSTs have also been shown to be active toward endogenous toxic compounds such as 4-hydroxynonenal, which is highly reactive and can affect protein function [6]. The catalytic efficiency of GSTD1 is of the same order as the catalytic efficiency of mammalian GSTs described as specialized in this function [6]. In addition, isomerase activity was recently described for GSTE14, previously shown to be involved in ecdysone biosynthesis, which is essential for molting (ecdysis) [7,8]. GSTE14 presents steroid double-bond isomerase activity, which is only 50-fold lower than the steroid double-bond isomerase activity of the most efficient mammalian GSTs [9]. In addition to this specific function of GSTE14, in general, the activity of the Delta and Epsilon classes is involved in resistance to synthetic insecticides such as organophosphate or natural insecticides such as isothiocyanates produced by plants [10,11,12,13,14,15,16,17]. Overexpression of GSTE7 in *D. melanogaster* provides significant protection against isothiocyanates found in cruciferous species [18]. However, insecticide-resistant insect strains show elevated expression of primarily Delta and Epsilon GSTs rather than changes in the expression of a particular GST gene.

Because of the redundancy of GST Epsilon and Delta functions, some of these GSTs have potentially evolved to support new functions in different insect species. Moreover, some organs are more prone to be in contact with xenobiotics than others. Thus, one can wonder if GST expression differs depending on the organ. GST exhibiting a specific function could be expressed in a specific organ, whereas GST with a generalist detoxification function could have ubiquitous expression in the body. This work aims to decipher the GST Delta and Epsilon amino acid sequences and their localization in different *D. melanogaster* organs using quantitative polymerase chain reaction (qPCR) analysis. After an analysis of the Delta and Epsilon GST amino acid sequences, the expression level of the corresponding coding mRNA will be measured in the different organs of the fly. Then, this localization will be discussed with respect to the knowledge of its biological function.

## 2. Materials and Methods

### 2.1. Sequence Alignment of Delta and Epsilon GSTs from Drosophila melanogaster

The amino acid GST sequences were downloaded from the flybase website (https://flybase.org/ (accessed on 20 May 2022)) [19]. Sequences were aligned using BioEdit software (https://bioedit.software.informer.com/ (accessed on 20 May 2022)).

### 2.2. Bioinformatics and Phylogeny

For the phylogeny, amino acid sequences from the Delta and Epsilon clades were retrieved from flybase (https://flybase.org/ (accessed on 20 May 2022)) and aligned using the online software MAFFT version 7 using the progressive method G-INS-I [20]. Phylogenetic reconstruction was performed by applying the maximum likelihood method [21]. The substitution model LG  +  I  +  G  +  F was suggested as the best-fitting model for protein evolution by the SMS server [22]. Trees were created by using the ATGC PhyML 3.0 bioinformatic platform. For tree topology improvement, both SPR (Subtree-Pruning-Regrafting) and NNI (Nearest-Neighbor-Interchange) methods were applied. The Gamma shape parameter and the proportion of invariable sites were calculated by the SMS server [23]. Substitution rate categories were set as 4. Node support was estimated using a hierarchical likelihood-ratio test [24,25]. Tree editing was performed with Figtree software version 3 (http://tree.bio.ed.ac.uk/software/figtree/ (accessed on 20 May 2022)).

### 2.3. Drosophila Strain and Rearing Conditions

W^1118^ wild-type male flies were reared on standard yeast/cornmeal/agar medium in a humidified, temperature-controlled incubator at 25 °C on a 12 h light:12 h dark cycle. Males were chosen to limit the variability compared to the females.

### 2.4. Tissue Collection, RNA Extraction and Real-Time (RT)–qPCR

Antennae, palps, labellum, heads without chemosensory appendages, guts (foregut, midgut and hindgut), and the carcass (thorax without wings and legs, and abdomens deprived of gut) were collected from 5-day-old adult males. In detail, after separating the heads from the bodies, the labella were dissected manually and the heads that still have the antennae and maxillary palps were placed in liquid nitrogen. Frozen heads were placed on a device comprising a tube surmounted by meshes and vortexed to separate olfactory appendages from the head. Finally, the gut was dissected from the rest of the body without the wings and legs. For the appendage, head without appendages, gut and carcass RNA extraction, we used about 1000 chemosensory appendages, 10 heads, 20 guts and 15 carcasses, respectively. Total RNA was extracted using TRIzol reagent (Invitrogen, Waltham, MA, USA) and treated with RNase-free DNAse (Thermo Fisher, Waltham, MA, USA) to avoid genomic DNA contamination. Total RNA was reverse-transcribed using the iScript cDNA Synthesis Kit (BioRad, Hercules, CA, USA). The qPCRs were carried out on a Step One real-time PCR system (Thermo Fisher, Waltham, MA, USA) using the Takyon Low Rox SYBR (Eurogentec, Seraing, Belgium). Each reaction was performed in triplicate with at least three independent biological replicates, and all results were normalized to the Rpl18, Rps20 and Rpl32 mRNA levels and calculated using the Pfaffl method [26]. Specific primers were designed using Primer-BLAST [27] and are shown in Appendix A.

## 3. Results

### 3.1. Delta and Epsilon GST Sequence Analysis

Sequence alignment of Delta and Epsilon GSTs reveals the presence of some conserved amino acids (yellow and blue in Figure 1). GSTs display different catalytic amino acids in the active site, Cys, Ser or Tyr, depending on the class. Delta and Epsilon harbor a Ser amino acid (position 33 on Figure 1) involved in the stabilization of the thiolate of the glutathione moiety. All Epsilon GSTs from *D. melanogaster* harbor the Ser residue, contrary to the Delta class. Indeed, GSTD3 has a shorter sequence with a missing N-terminal part, explaining the absence of Ser. GSTD3 did not present any activity with the tested substrates [5]. It cannot be excluded that GSTD3 has a conserved ligand in function consisting of carrying molecules without activity. Interestingly, GSTD2, GSTD5 and GSTD7, which do not contain a Ser residue, are all catalytically active [5,28]. X-ray structure resolution of GSTD2 did not show another residue potentially replacing the Ser; the activity was thus proposed to be supported by other residues within the active site allowing the deprotonation of the glutathione sulfur required for catalysis [28]. Additionally, the GSTD11A sequences bear an additional sequence corresponding to GSTD11B that includes an N-terminal extension in the GST11A sequence. This extension was not predicted as a signal peptide using the SignalP website (https://services.healthtech.dtu.dk/service.php?SignalP-5.0 (accessed on 12 May 2022)); however, it cannot be ruled out that it is a noncanonical signal peptide. Delta and Epsilon Drosophila GSTs also share common features in their amino acid sequences. First, the glutathione-binding site consists of residues from the thioredoxin fold, including the conserved cis-Pro of the β3 strand and the first two residues N-terminal (Glu/Asp-Ser) to the α3 helix, also including a His residue from a conserved beta turn that joins helix α2 and the β3 strand. Second, the α3 helix and α4 helix are fairly conserved in both classes due to their importance for dimerization. The conserved residues further include those involved in dimerization motifs, such as the “lock-and-key” residue, which is a Tyr/Phe for the Delta GST class (named ‘clasp’) and a His for the Epsilon class (named ‘wafer’) [29,30,31,32,33,34,35,36,37]. They participate in the stabilization of the dimer by hydrophobic effects [34]. Third, an N-capping box found in all GSTs involved in GST folding is also present in the N-terminus of the α6 helix in both the Delta and Epsilon classes. This motif is important for stabilization of the central α6 helix [38]. The identification of the hydrophobic site involved in substrate recognition located from the α4, α6 and α9 helices remains challenging since it most often involves non conserved hydrophobic residues, explaining why no conserved residue was identified for this hydrophobic site.

### 3.2. Phylogenic Analysis

The Delta and Epsilon classes specifically found in arthropods appear with a large number of members in *D. melanogaster* (25 of the 42 GSTs in total). Phylogenic analysis of the Delta and Epsilon GST protein sequences allowed us to observe two separate groups, one for each class (Figure 2). The Delta classes appear organized into two main subgroups. One of these subgroups is formed by GSTD11A and GSTD11B. These GSTs differ only by an additional N-terminal sequence for GSTD11B. The group formed by GSTD11A and GSTD11B was probably separated early during the evolutionary process of the Delta class. The other Delta GSTs are more similar, suggesting a recent evolution compared to the Epsilon GSTs. However, the other Delta GSTs can be separated into three groups, one with only GSTD8, another one including GSTD1, GSTD9 and GSD10, and a larger third group with all the remaining Delta. This last large group appeared very recently and consequently was one of the most important drivers for the recent diversification of the Delta class in *D. melanogaster*. Epsilon GSTs are organized into three subgroups: one resulting from an ancient separation, including GSTE11, GSTE12, GSTE13 and GSTE14. The two other groups of Epsilon were formed a second time. The diversification of the last two groups was scaled on the largest period of time compared to the Delta GSTs. Compared with the other Delta GSTs, GSTE13 and GSTE14 have higher evolutionary rates, as mirrored by the relatively long branch lengths in the Delta clade of the phylogeny.

### 3.3. Expression of mRNA Encoding GST

To support the potential functions of the 25 Delta and Epsilon GSTs, the transcript abundance of the corresponding genes was measured in the antennae, labellum, head, gut and body (Figure 3). Sensory organs (antennae and labellum), as well as the gut, appear to be the organs most exposed to exogenous molecules. In view of the major role of GSTs in the detoxification process, this study focused on these tissues. The expression levels differed between organs and GSTs. In general, GSTD5 and GSTE5 were not expressed in the tested tissues. Different primers were tested without success; however, experimental bias cannot be excluded. mRNA encoding GSTD5 and GSTE5 was previously found using RNA sequencing in a complementary approach [39]. Consequently, GSTD5 and GSTE5 were not included in the results presented here. GSTD6, GSTD7, and GSTE1 were expressed at a low level in the tested tissues compared to other GSTs, with no specific expression in any of the tested conditions. Some GSTs appeared to be specifically expressed in certain tissues, such as GSTE4 in antennae, and others were similarly expressed in all the tested tissues, such as GSTD1, GSTD9, GSTE9 and GSTE13. The expression level of GSTs is not associated with the sequence proximity; additionally, no relationship was found between expression specificity and clustering in the phylogenetic tree.

#### 3.3.1. Antennae

GSTE4, GSTD1, GSTE12 and GSTD4 were the most highly expressed in antennae. Some of them are highly specific to antennae compared to the other organs tested (Figure 3). For example, mRNA-encoding GSTE4 transcripts are 50 times more highly expressed in antennae than in the head, which is the second most GSTE4-expressing tissue. GSTD11 did not appear to be most expressed in the antennae, but like GSTE4, this gene is specifically expressed in this organ. GSTD11 is 121 times more expressed in the antennae compared to the gut, the second place where it is the most expressed. The specific expression of GSTE4 and GSTD11 in the antennae suggests a potential role of these GSTs in olfaction. Furthermore, GSTD1, GSTE12 and GSTD4 are highly expressed in the antennae but also show high expression levels in several other organs, suggesting a more general function.

#### 3.3.2. Labellum

GSTD4, GSTD1 and GSTE12 appeared to be most highly expressed in the proboscis (Figure 3). These three GSTs were already among the most expressed in the antennae; however, they all appear at a higher expression level in the proboscis. None of these three GSTs appear to be highly specific to this tissue. GSTD4 is only 2.7 times more highly expressed in the labellum than in the gut. A comparison of the expression of all GSTs in all organs shows that the highest GST expression is GSTD4 in the labellum. GSTD1 and GSTE12 are also highly expressed in this tissue. These three proteins are expressed in all organs.

#### 3.3.3. Gut, Head and Carcass

As in the labellum, the three most expressed GSTs in the gut, head without appendages, and carcass are GSTD4, GSTE12 and GSTD1. However, they are expressed at a lower level compared to the labellum, and the lowest expression is detected in the head (Figure 3). Interestingly, drastic differences can be observed for some GSTs in relation to these three tissues. GSTE14 is specifically expressed in the head, 50 times more compared to any other location tested, and GST E10 is mostly expressed in the gut and carcass compared to other locations. Some GSTs appear less expressed in specific organs. GSTD7 is on average 10 times less expressed in the head compared to other locations. The same observation is made for GSTE4 and GSTE14 in the carcass compared to any other location tested.

## 4. Discussion

GSTs are ubiquitous enzymes; six classes are found in insects, including the Delta and Epsilon classes, which are insect-specific. Delta and Epsilon GSTs have been shown to be involved in different functions, such as insecticide resistance and ecdysone biosynthesis. Phylogenetic analysis revealed the diversity of the Delta and Epsilon classes, supported by the different subgroups emerging within these classes in the phylogenetic tree. Sequence analysis additionally shows amino acid differences in the sequence forming the active site as well as the amino acid sequence involved in the dimerization motif. The length of the GST sequences also diverges between sequences. GSTD11b appears with an additional N-terminal part, and GSTD3 appears shorter. Additionally, in this study, qPCR analysis of mRNA-encoding GST Delta and Epsilon was performed. First, the results on Delta and Epsilon GSTs support a previous RNA sequencing analysis conducted on other organs [39]. Both approaches are complementary and reveal similar results regarding the high expression levels of GSTD1, GSTD11, and GSTE4 in antennae [39]. Additionally, the qPCR study revealed a high expression level of mRNA-encoding GSTD4 in the antennae and labellum. The specific expression of GSTE4 and GDTD11 in antennae was confirmed in the qPCR study by analyzing their expression levels in other organs, such as the labellum. This result suggests a potential role of these GSTs in the antennae involved in fly olfaction. The qPCR study provides new evidence of the high expression of some GSTs in this particular organ, such as GSTE12, GSTD1 and GSTD4. The absence of catalytic activity previously reported for GSTD3 and supported by the shorter N-terminal part of the sequence consequently removing the catalytic site raises questions about the function of this GST in view of its substantial level of expression in all organs tested. Additionally, mRNA-coding GSTD3 expression was previously shown to drastically change during development, with a higher expression in the central nervous system of the pupae [39]. GSTs have been previously shown to have a ligandin function involved in the transport of molecules. Consequently, if this function is conserved in GSTD3, this may explain the conservation of its expression in the different organs tested. The specific expression of GSTE14 in the head highlights a potential link of the specific isomerase activity of this particular GST [9] within this organ. Conversely, the very low specific expression of GSTE14 in the carcass raises the question of whether it is supported by a biological explanation, e.g., whether the isomerase activity disrupts certain reactions. GSTD1, which has previously been shown to be important for insecticide resistance [10,12,13,14,15,16,17,28], was found to be highly expressed in all body sites tested. This result seems to be consistent with this function; however, we cannot exclude other general functions. Accurate localization within the tissue will be valuable. An omega GST was shown to be expressed in the fly eyes and shown to be linked to the eye color [40]. Approaches such as immunohistochemistry are not appropriate to study GST member localization; indeed, antibodies against GSTs from a same class present cross-reactivity [41,42]. Therefore, in situ hybridization should be more adapted in future studies to decipher GST localization within the tissues. GST expression during the developmental stage appears to be of major interest, as pointed out by GSTD3. GSTD5 mRNA expression was already identified to significantly increase in the antennae of virgin females compared to mated females [39]. Consequently, a complete sex-dependent GST expression analysis will also be valuable to better understand the role of insect GSTs.

## Figures and Tables

**Figure 1 insects-13-00612-f001:**
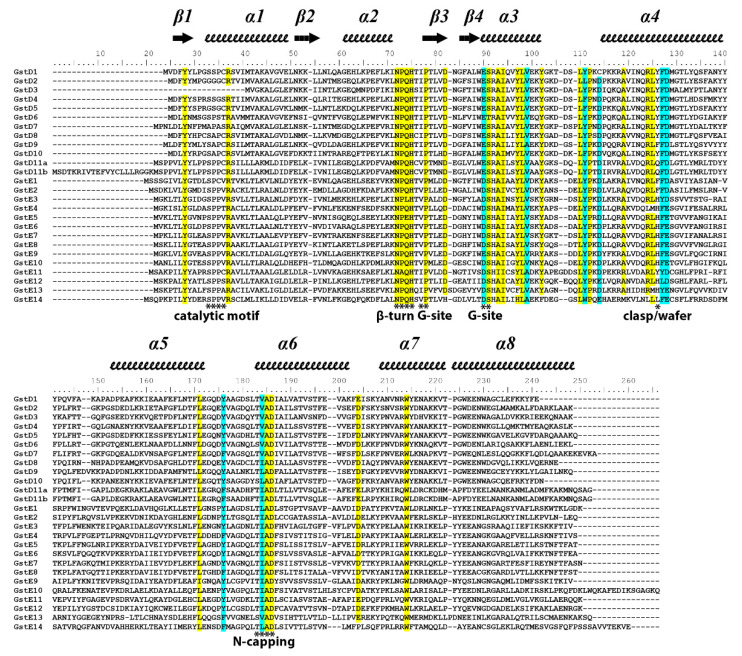
Sequence alignment of Delta and Epsilon GSTs from *Drosophila melanogaster*. Secondary structures have been inferred from the X-ray structure of DmGSTD2 (5F0G, [28]) and reported above the alignment. Conserved sites are annotated below the alignment with a star indicating the involved amino acids.

**Figure 2 insects-13-00612-f002:**
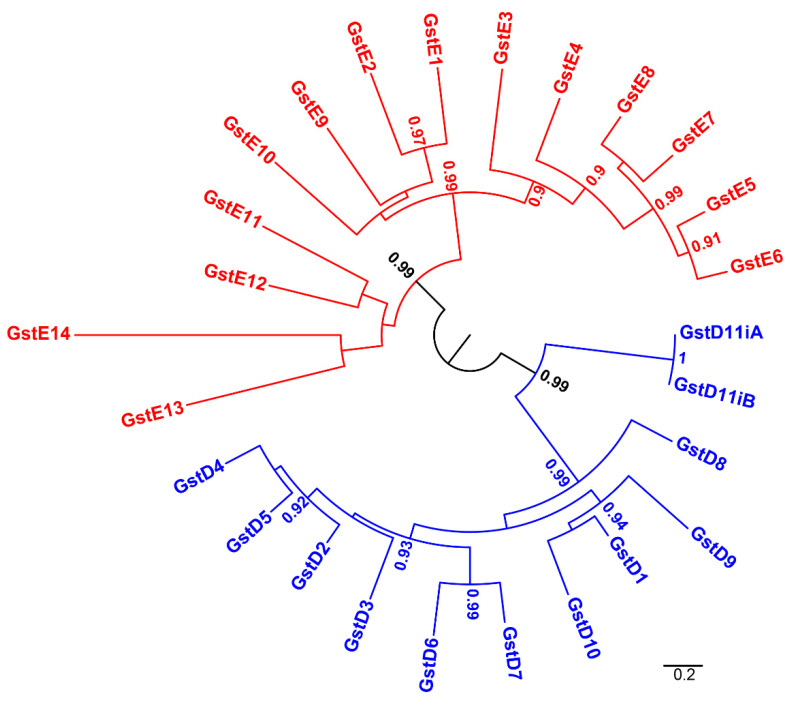
Evolutionary relationships of Delta and Epsilon GSTs of *Drosophila melanogaster*. Phylogeny was created from full-length protein sequences applying the maximum likelihood method [21]. Dots represent branch support values based on the fast likelihood method, aLRT  ≥  0.9; aLRT  <  0.9 were discarded (aLRT corresponds to the approximate Likelihood-Ratio Test). The branch length corresponds to the number of amino acid substitutions, and the scale bar indicates the average number of amino acid substitutions per residue. Delta and Epsilon GST are represented in blue and red, respectively.

**Figure 3 insects-13-00612-f003:**
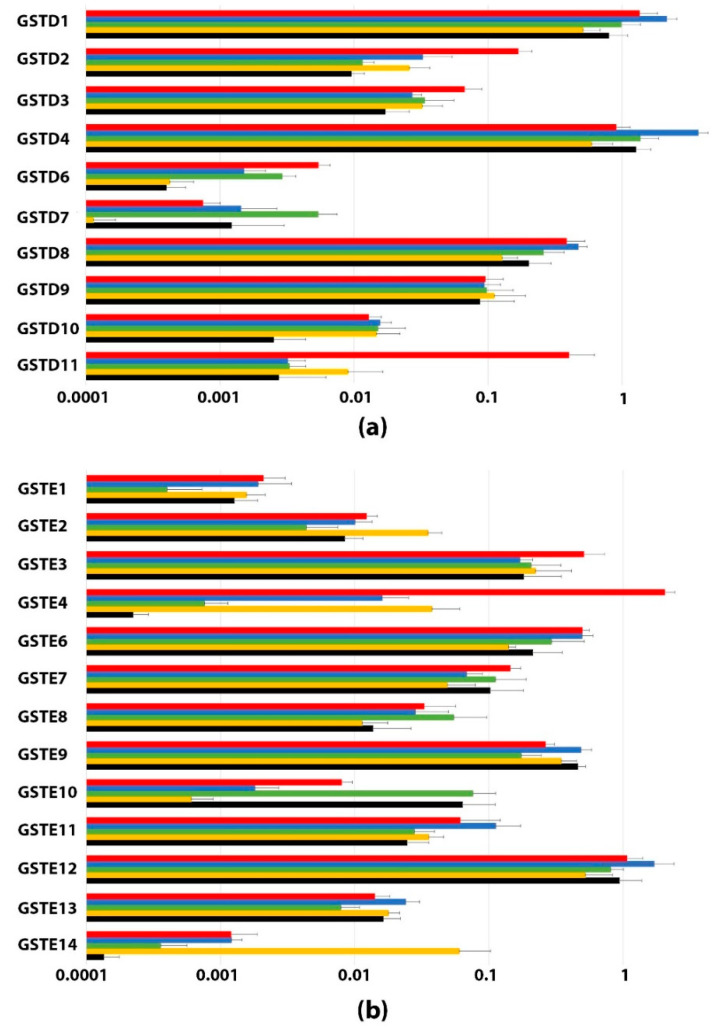
Expression levels of Delta and Epsilon GSTs from *D. melanogaster* in various tissues. (**a**) GST Delta and (**b**) GST Epsilon. qRT-PCR was performed on olfactory organs (antennae and maxillary palps in red), taste organs (labellum in blue), the gut in green, the head in yellow (without chemosensory appendages) and the carcass in black for GST Delta (**a**) and GST Epsilon (**b**). All data corresponding to the normalized expression level are plotted as the mean ± SD of triplicate biological samples on a log scale.

## Data Availability

Data supporting reported results are all included.

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
