# Peer review of "Expression Patterns of Drosophila Melanogaster Glutathione Transferases"

_insects, 2022, doi:10.3390/insects13070612_

Round 1
Reviewer 1 Report
This paper pretends to establish a link between GSTs amino acid sequences and their expression pattern in different Drosophila tissues.
Major Points:
However, I have some concern about the take home message of this paper. The authors accurately define a phylogenetic pattern of GSTs and only confirm previous RNA-seq data by qRT-PCR.
Anyway, in my opinion they should refer to the last release of Drosophila genome sequencing for their analysis:
- Hoskins RA, Carlson JW, Wan KH, Park S, Mendez I, Galle SE, Booth B, Pfeiffer BD, George RA, Svirskas R, Krzywinski M, Schein J, Accardo MC, Damia E, Messina G, Demakova OV, Andreyeva EN, Lidiya Boldyreva V, Moore R, Marra M, Carvalho AB, Villasante A, Dimitri P, Zhimulev IF, Rubin GM, Karpen GH, Celniker SE. The Release 6 Drosophila melanogaster reference genome. Genome Res, 2015. Jan 14. pii: gr.185579.114
This reference should be cited at the beginning of the “Material and Methods” section of the paper.
Moreover, it becomes clear only at row 97 that authors refers to amino acid sequences of GSTs. It should be declared earlier in the paper.
Minor Points:
Row 58: “(GSTD an GSTE)” – I would change in “(GSTD an GSTE, respectively)”
Row 86: “This work aims to decipher the 86 GST Delta and Epsilon sequences...” – which kind of sequence? DNA, RNA or amino acids?
Row 89: same of row 86
Row 94: It is not clear which kind of GST sequences were downloaded from Flybase
Row 110: “w1118” should be changed in “w1118”. It is clear why males are used instead females, but it should be explained
Author Response
Dear reviewer,
Thank you for your positive come back.
We have taken all the comments into account (see the detailed point-by-point below). We are pleased that our work is recognized of interest by the three reviewers and seen as an interesting contribution to the field.
Reviewer 1
This paper pretends to establish a link between GSTs amino acid sequences and their expression pattern in different Drosophila tissues.
Major Points:
However, I have some concern about the take home message of this paper. The authors accurately define a phylogenetic pattern of GSTs and only confirm previous RNA-seq data by qRT-PCR.
Anyway, in my opinion they should refer to the last release of Drosophila genome sequencing for their analysis:
- Hoskins RA, Carlson JW, Wan KH, Park S, Mendez I, Galle SE, Booth B, Pfeiffer BD, George RA, Svirskas R, Krzywinski M, Schein J, Accardo MC, Damia E, Messina G, Demakova OV, Andreyeva EN, Lidiya Boldyreva V, Moore R, Marra M, Carvalho AB, Villasante A, Dimitri P, Zhimulev IF, Rubin GM, Karpen GH, Celniker SE. The Release 6 Drosophila melanogaster reference genome. Genome Res, 2015. Jan 14. pii: gr.185579.114
This reference should be cited at the beginning of the “Material and Methods” section of the paper.
It is done
Moreover, it becomes clear only at row 97 that authors refers to amino acid sequences of GSTs. It should be declared earlier in the paper.
It is done, we have adapted the end of the introduction to clarify this.
Minor Points:
Row 58: “(GSTD an GSTE)” – I would change in “(GSTD an GSTE, respectively)”
It is done
Row 86: “This work aims to decipher the 86 GST Delta and Epsilon sequences...” – which kind of sequence? DNA, RNA or amino acids?
We added amino acids to clarify.
Row 89: same of row 86
We added amino acids to clarify.
Row 94: It is not clear which kind of GST sequences were downloaded from Flybase
We added amino acids to clarify.
Row 110: “w1118” should be changed in “w1118”. It is clear why males are used instead females, but it should be explained
It is done. The explanation was added in MM section.
Reviewer 2 Report
The paper by Gonis et al. is generally well written and experiments are well designed. I just have few minor comments which will help to improve the clarity and readability of this manuscript.
Required minor changes:
1. Readers could benefit if some description on GST classification system in all living organisms is outlined in the introduction section.
2. Lines 53-54: Rephrase this sentence for improved clarity. Does Apis mellifera only have one Delta family GST gene?
3. Lines 75-76: Isothiocyanates are not synthetic insecticides like the Organophosphate class insecticides. Rephrase this sentence to provide accurate information on isothiocynates.
4. Section 2.3 or 2.4: Include justification on why adult males flies were used for mRNA expression experiments.
5. It is not clear if the information provided below each figure (1-3) is a part of the text or the caption.
6. Double check if appropriate references are cited in section 3.2.
7. Lines 203-205: I did not see the correlation data and associated statistics presented in this paper.
8. Lines 242-245: This description is not required. Is appears as if these lines are comments made by a reviewer on an initial version of this manuscript.
Author Response
Dear reviewer 2,
Reviewer 2
Thank you for your positive come back.
We have taken all the comments into account (see the detailed point-by-point below). We are pleased that our work is recognized of interest by the three reviewers and seen as an interesting contribution to the field.
Comments and Suggestions for Authors
The paper by Gonis et al. is generally well written and experiments are well designed. I just have few minor comments which will help to improve the clarity and readability of this manuscript.
Required minor changes:
- Readers could benefit if some description on GST classification system in all living organisms is outlined in the introduction section.
A short explanation was added in the Introduction : « The classes are designated by the names of the Greek letters, generally a class contain several GSTs which are numbered using Arabic numerals. »
- Lines 53-54: Rephrase this sentence for improved clarity. Does Apis mellifera only have one Delta family GST gene?
The new sentence : « For instance, Apis mellifera has only one GST member in the Delta class and no GST member in the Epsilon class. »
- Lines 75-76: Isothiocyanates are not synthetic insecticides like the Organophosphate class insecticides. Rephrase this sentence to provide accurate information on isothiocynates.
The new sentence : « In addition to this specific function of GSTE14, in general, the activity of the Delta and Epsilon classes is involved in resistance to synthetic insecticides such as organophosphate or natural insecticides such as isothiocyanates produced by plants »
- Section 2.3 or 2.4: Include justification on why adult males flies were used for mRNA expression experiments.
The explanation was added in MM section.
- It is not clear if the information provided below each figure (1-3) is a part of the text or the caption.
It was changed to make it clearer
- Double check if appropriate references are cited in section 3.2.
A reference was added
- Lines 203-205: I did not see the correlation data and associated statistics presented in this paper.
Sorry, the word « correlation » is not used in a statistical analysis way. We changed the sentence for « The expression level of GSTs is not associated with the sequence proximity; additionally, any relationship was found between expression specificity and clustering in the phylogenetic tree »
- Lines 242-245: This description is not required. Is appears as if these lines are comments made by a reviewer on an initial version of this manuscript.
Right, we remove these comments.
Reviewer 3 Report
This manuscript titled “Expression patterns of Drosophila melanogaster glutathione transferases ” describes a study of the insect specific delta and epsilon subgroups of Glutathione transferases that are composed of 25 GSTs in Drosophila. The authors compared protein sequences between each group and revealed that amino acids that are essential for structure stability and dimerization are conserved through all the GSTs. Further qPCR quantitive analysis from different tissues in Drosophila revealed some GSTs are tissue specific expressed, suggesting a functional diversity in these GSTs.
Overall, this paper contains some interesting findings. However, the data from this paper are insufficient to support the conclusion, and the following questions need to be addressed for further evaluation.
1. qPCR analysis is effective but the detailed expression pattern is missing and often contain false positive signals due to its sensitivity. In-situ hybridization or immune-staining (only if the antibodies are available) with specific anti-bodies are necessary for further confirm the results from qPCR and will provide more comprehensive understanding of cellular location of the GSTs.
2. Some key information was missing in the material and methods chapter. For example, how were the samples collected for qPCR? How many tissues are used in each experiment? How many times are qPCR repeated?
3. Is there any endogenous control to make sure that the samples collected for qPCR were not contaminated by other tissues or body parts?
4. Reference missing: line 186: what is aLRT? line 275-276.
5. Discussion is not adequate: specific expression of GSTS is interesting, need more discussion to link from expression to potential function. For example, the predominant expression of mRNA in heads is from photoreceptor. The head specific expressed GSTE14 suggests that it should be function in visual system.
6. This is not required but would be interesting to figure out or discussed in the paper. (1) are there any distinct expressions between male and female flies? (2) is there any expression change during the developmental stages?
Author Response
Dear reviewer 3,
Thank you for your positive come back.
We have taken all the comments into account (see the detailed point-by-point below). We are pleased that our work is recognized of interest by the three reviewers and seen as an interesting contribution to the field.
Comments and Suggestions for Authors
This manuscript titled “Expression patterns of Drosophila melanogaster glutathione transferases ” describes a study of the insect specific delta and epsilon subgroups of Glutathione transferases that are composed of 25 GSTs in Drosophila. The authors compared protein sequences between each group and revealed that amino acids that are essential for structure stability and dimerization are conserved through all the GSTs. Further qPCR quantitive analysis from different tissues in Drosophila revealed some GSTs are tissue specific expressed, suggesting a functional diversity in these GSTs.
Overall, this paper contains some interesting findings. However, the data from this paper are insufficient to support the conclusion, and the following questions need to be addressed for further evaluation.
- qPCR analysis is effective but the detailed expression pattern is missing and often contain false positive signals due to its sensitivity. In-situ hybridization or immune-staining (only if the antibodies are available) with specific anti-bodies are necessary for further confirm the results from qPCR and will provide more comprehensive understanding of cellular location of the GSTs.
Thanks for your suggestions. We fully agree that immunostaining will be valuable. Unfortunately, one of the characteristics of GST belonging a same class is the antibodies cross-reactivity. Consequently, this technic is not appropriate to distinguish the GST members of the Delta or Epsilon class. In-situ hybridization showing the RNA expression within the different tissues will complete in the future the knowledge on GSTs based on this study setting one of the first stone on this question.
- Some key information was missing in the material and methods chapter. For example, how were the samples collected for qPCR? How many tissues are used in each experiment? How many times are qPCR repeated?
We adapt the MM to your suggestions and answer all your questions in the new MM.
- Is there any endogenous control to make sure that the samples collected for qPCR were not contaminated by other tissues or body parts?
We used a very rigorous and accurate dissection protocol that significantly reduces the problem of contamination between tissues and we carried out three tissue dissections separately. In addition, as we mentioned in the discussion, our results are similar to those obtained in a previous RNA seq analysis.
- Reference missing: line 186: what is aLRT? line 275-276.
aLRT correspond to the approximate Likelihood-Ratio Test, we added this information in the paper.
The reference was added.
- Discussion is not adequate: specific expression of GSTS is interesting, need more discussion to link from expression to potential function. For example, the predominant expression of mRNA in heads is from photoreceptor. The head specific expressed GSTE14 suggests that it should be function in visual system.
We adapted the end of the discussion to take your remark into account, and open the question of a more accurate localization, valuable for example in the case of the omega GST member involved in fly eye color.
- This is not required but would be interesting to figure out or discussed in the paper. (1) are there any distinct expressions between male and female flies? (2) is there any expression change during the developmental stages?
In this paper, we did not compare the males and females, but we already knew that GSTs are not expressed at the same level between the males and females. Additionally, their expression seems also to be different during the development opening numerous questions for the future.
Round 2
Reviewer 3 Report
I'm pleased to see that most of my suggestions/questions are properly addressed, and don't have any further questions except the following one.
1. as mentioned in the 1st round review report, GST expression in different sex and developmental stages should be addressed in the discussion chapter.
Author Response
Dear reviewer 3;
Thanks, we took in account your last suggestion and modify the discussion consequently,
Best regards